# Prevalence of Eating Disorders and Disordered Eating Behaviours amongst Adolescents and Young Adults in Saudi Arabia: A Systematic Review

**DOI:** 10.3390/nu15214643

**Published:** 2023-11-01

**Authors:** Azzah Alsheweir, Elizabeth Goyder, Ghadah Alnooh, Samantha J. Caton

**Affiliations:** 1Sheffield Centre for Health & Related Research (SCHARR), School of Medicine & Population Health, University of Sheffield, Sheffield S1 4DA, UK; e.goyder@sheffield.ac.uk (E.G.); s.caton@sheffield.ac.uk (S.J.C.); 2Department of Community Health Sciences, College of Applied Medical Sciences, King Saud University, Riyadh 145111, Saudi Arabia; 3Centre for Assistive Technology and Connected Healthcare, Sheffield Centre for Health & Related Research (SCHARR), School of Medicine & Population Health, University of Sheffield, Sheffield S1 4DA, UK; gsaalnooh1@sheffield.ac.uk; 4Department of Health Sciences, College of Health and Rehabilitation Sciences, Princess Nourah Bint Abdulrahman University, Riyadh 11671, Saudi Arabia

**Keywords:** eating disorders, disordered eating behaviours, weight, body image, EAT-26, EDI-DT, EDE-Q, SCOFF, Saudi Arabia

## Abstract

Eating disorders (EDs) and disordered eating behaviours (DEBs) are significant health problems related to body image and weight dissatisfaction among adolescents and young adults worldwide. During this stage of sensitive development, these behaviours may hinder the optimal growth process and can consequently lead to wasting, stunting, and micronutrient deficiencies. However, there is a limited amount of literature on DEB among Arab populations, particularly in Saudi Arabia. This review aims to assess the prevalence of EDs/DEB and to develop a clear understanding of the epidemiology of such conditions among adolescents and youth in Saudi Arabia. Five databases were systematically searched and 14 papers met the inclusion criteria. The Eating Attitudes Test (EAT-26) was the predominant tool used for evaluating aberrant eating behaviours, indicating a high prevalence of EDs/DEB ranging from 10.2% to 48.1%. The highest prevalence of EDs/DEB was reported in the eastern region ranging from 29.4% to 65.5%. In terms of age and sex, the prevalence was higher among older students compared to younger school students and males reported more Eds/DEB compared to female students. These disorders are found to be prevalent in Saudi Arabia, and the risk of developing such conditions is high across the whole age range.

## 1. Introduction

Eating disorders (EDs) appear to be one of the most significant problems concerning adolescents and young people [1]. According to the Diagnostic Statistical Manual of Mental Disorders (DSM-V) and the International Classification of Diseases and Related Health Problems (ICD-11), EDs are characterised by troubled eating habits and detrimental practices of weight control [2,3]. They are ranked as the third most common pathological diagnoses in adolescents worldwide, following asthma and obesity [4]. The WHO and the American Psychiatric Association (APA) classified anorexia nervosa (AN), bulimia nervosa (BN), and binge eating disorder (BED) as the most common EDs globally [5,6,7]. AN is defined as a lack of desire to eat and loss of appetite combined with compensatory behaviours towards thinness and fear of gaining weight [7]. A recent review of epidemiological studies confirmed that the incidence of AN is increasing in young individuals worldwide, particularly those who are below 15 years of age, with a peak age of 13–18 years at onset [8,9]. The lifetime prevalence of AN is 4% among female populations and 0.3% among male populations. AN has the highest mortality rate among all psychiatric illnesses [8,10]. BN is delineated by recurrent binge eating, followed by purging behaviours (self-induced vomiting and laxative or diuretic use) [7,11]. The lifetime prevalence of BN is up to 3% among female populations and over 1% among male populations, with symptoms increasing among girls from 14 to 16 years of age and among boys from 16 years of age onwards [8,12]. BED is described as persistent binge eating episodes combined with feelings of shame and depression, rather than purging or other compensatory behaviours (diet pill intake, intensive exercise, and diet restriction) [11,13]. In the former DSM-IV, BED was counted as a significant diagnosis categorised under eating disorders not otherwise specified [14]. However, in the current version of the manual (DSM-V), BED is considered a distinct diagnosis and a substantial disorder along with anorexia and bulimia [5,15]. Binge eating and other behaviours in adolescents have an estimated lifetime prevalence of 4.8% [16]. Although girls account for the majority of cases of EDs, 10% of cases have been reported by boys over the last few years [17].

Disordered eating behaviours (DEBs) are significant behaviours and attitudes from the ED spectrum [18,19]. These disordered behaviours and negative attitudes towards food can be practised by individuals without being diagnosed with a specific ED [19]. Neumark-Sztainer et al. explained that behaviours such as binge eating, purging, fasting, and using laxatives are parallel to the behaviours manifested in clinical EDs [20]. The literature has focussed on EDs more than on DEB, since EDs can be diagnosed by a multidisciplinary team under clinical conditions through physical and psychological screening [7,21]. The DSM-V and ICD-11, which are validated diagnostic tools, can detect the prevalence and severity of such disorders [3,22]. By contrast, detecting DEB is not feasible, as clinical professionals and even individuals are not aware of the symptoms, and treatment plans are lacking [21].

This review highlights the current knowledge on the prevalence of EDs and DEB, with variation in prevalence associated with sex, age, region, and types of tools used. DEBs are found to be a major health concern related to body image and weight dissatisfaction in adolescents [20]. These behaviours have been reported in both sexes among various cultural, ethnic, racial, and socioeconomic backgrounds [23]. However, female adolescents tend to have higher rates of DEB because they are more concerned about their body shape and physical appearance than males [24]. In the US and Europe, approximately 10–20% of female adolescents engage in DEB, and estimates may reach 40% for certain behaviours [20,25]. As adolescents experience remarkable growth and increased nutrient requirements, failure to meet such requirements can impair normal growth and sexual maturation [26,27,28]. Energy restriction can be problematic for adolescents, as it may lead to binge eating and an increased risk of EDs [29].

Podar and Allik (2009) indicated that the highest proportion of DEB was detected in mid-to-late adolescence (15–21 years) [23]. According to the current literature, EDs often appear in adolescence or early adulthood, with reported median onset ages between 12 and 22 years [30,31]. It is therefore highly plausible that the transition into and out of adolescence is related to an increased risk of EDs [32]. Yearly estimates indicated that for both males and females, the highest mean prevalence of EDs occurred at the age of 21 years, with the majority of cases (95%) occurring by the age of 25 for the first time [33].

The literature indicated that EDs and pathological concerns with body image and food are present in Western Europe and North America [1]. However, these behaviours are also documented in other regions in Asia [34,35]. In Taiwan, results revealed that more than one-third (43.2%) of college students tend to have a higher prevalence of developing an ED [36]. A review from Singapore confirmed a marked increase in the prevalence of EDs over the past ten years, compared to older studies, and linking it to Westernisation and body dissatisfaction, both of which appear to be frequent among Singaporean Chinese schoolgirls [37]. The Middle East and North Africa (MENA) countries have widely documented DEBs among adolescents and young adults [38,39,40,41]. These countries are categorised as emerging economic countries, considering that they are experiencing significant transitions in culture, diet, and sex roles [34,42]. A study assessed eating attitudes among adolescents aged 15–18 years (n = 4698) as a part of the ARAB Eating Among Teens project in seven cities in Arab countries (Algeria, Jordan, Kuwait, Libya, Palestine, Syria, and UAE). It was concluded that DEB was highly prevalent in both male (13.8–47.3%) and female (16.2–42.7%) adolescents [43]. The prevalence of DE among Arab adolescents is comparable to that in adolescents from Western countries [44].

DEB could be attributed to the significant shifts in eating habits and lifestyles in Saudi Arabia [45,46]. Unhealthy eating habits and lack of physical activity can be detrimental to health during adolescence [47]. Moreover, Saudi adolescents are preoccupied with the thinness body ideal rather than the culturally preferred full body shape, which may also increase the possibility of EDs [24,48].

Epidemiological studies and systematic reviews have focussed on Western and non-Western distinctions in measuring the prevalence of EDs and DEB [1]. Furthermore, little epidemiological research has fully observed the association between urban region residence and disordered behaviour, confirming that the relationship between urbanisation and EDs is complex, not directly causal, and likely to vary between different contexts/populations [49].

No previous review has focussed on such disorders in populations in Saudi Arabia. In this regard, a systematic review is imperative to develop a clear understanding of the epidemiology of EDs and DEB among adolescents and young adults in Saudi Arabia. The objective of this review is to assess the prevalence of EDs and DEB, among Saudi adolescents and young adults, with an emphasis on variations related to sex, age, region, and types of tools used. As this is an exploratory study, these variables (sex, age, region) were emphasised to understand the demographic differences in relation to the prevalence of EDs/DEB in Saudi Arabia.

## 2. Materials and Methods

The Preferred Reporting Items for Systematic Reviews and Meta-Analyses (PRISMA) guidelines framework was used in conducting and reporting this systematic review [50]. This systematic review is registered and approved by Prospero (CRD42022359970). 

### 2.1. Data Retrieval Strategies

The search strategy for this systematic review was developed in consultation with an academic librarian. A comprehensive literature search was conducted in relevant databases to reduce the likelihood of missing evidence. Five online databases were selected for this review: MEDLINE via Ovid, Scopus, Web of Science, Psych Info, and ProQuest via ASSIA (1990–2022). Internet search engines were checked, and reference lists of selected papers were manually cross-checked to identify studies missed in the original search.

Keywords and search terms used in the databases search were related to eating disorders, disordered eating, adolescents, youth, and Saudi Arabia. MeSH terms, as well as appropriate synonyms, were applied. These terms were linked using the Boolean operators AND and OR. Search strategies were entered in February 2023 (Appendix A).

### 2.2. Inclusion and Exclusion Criteria

Studies were assessed and included based on the following eligibility criteria following the PICOs from the JBI manual for evidence synthesis [51] (Table 1). As clarified in the table, only studies that serve the aim of this systematic review are included, recruiting healthy adolescents from Saudi Arabia and defining the prevalence of EDs/DEB using a validated tool. As the inclusion criteria resulted in the identification of studies using a range of different methods to measure prevalence, the heterogeneity of studies meant that meta-analysis was not considered appropriate.

### 2.3. Screening and Data Extraction

Retrieved records were exported to EndNote 20 to remove duplicate studies. Studies were initially screened by title and then by abstract for relevance. Two reviewers (AA and GA) manually screened the titles and abstracts of the records retrieved from the search. References that did not meet the inclusion/exclusion criteria were withdrawn and the remaining references that met the search criteria went through a full-text assessment. Two authors (AA, GA) independently screened all potential full-articles, and the final decision was made by consensus. Any discrepancies were resolved by a third and a fourth reviewer (SC, EG). This systematic review is one of the outputs of AA’s Ph.D. project, supervised by SC and EG, who provided expert guidance based on extensive systematic review experience. Data from each record meeting the inclusion criteria were inserted into a table that has the following classifications: Author, City/Area, Study Design, Setting, Sample Size, Sex Distribution (F/M), Age Group (Mean Age ± SD), Screening Tool, Disorder Screened, and Prevalence (N and/or %). Disorder screened was identified according to the selected papers’ purpose which included ED, DEB, and DEA. However, DEB and DEA included the same behaviours.

### 2.4. Quality Assessment Tool

The quality of retrieved records was independently assessed and collectively verified by two authors (AA, GA) using the Joanna Briggs Institute (JBI) critical appraisal tool for cross-sectional studies [51]. Sections assessed include eight identified criteria: (Q1) inclusion in the sample, (Q2) study subjects and the setting, (Q3) methods of measuring the exposure, (Q4) criteria used for measurement of the condition, (Q5) confounding factors, (Q6) strategies to deal with confounding factors, (Q7) methods of measuring the outcomes, and (Q8) statistical analysis used. It is important to note that criteria five and six (Q5 & Q6) were rated as not applicable, as they are concerned with confounding factors and strategies to deal with confounding factors. After deleting Q5 and Q6, scores were calculated depending on 6 criteria, where a score of 1 is assigned to each criterion. A score of 5–6 indicates high-quality research, a score of 3–4 indicates medium quality, and 1–2 scores indicate low quality.

### 2.5. Data Synthesis

Since studies are employing similar tools (questionnaires) to ascertain the prevalence rate of such conditions, attention was placed on EDs/DEB. Data were analysed first by grouping studies by type of questionnaire: Eating Attitudes Test (EAT-26) [52,53], Eating Disorder Inventory—Drive for Thinness (EDI-DT) [54,55], Eating Disorder Examination—Questionnaire (EDE-Q) [56,57,58], and Sick, Control, One, Fat, Food questionnaire (SCOFF) [59,60,61]. Then, data were interpreted according to region, sex, and age. It is important to note that educational setting is used rather than age group per se because the main interest was identifying whether prevalence was different in school and college/university settings.

## 3. Results

### 3.1. Study Selection

The initial search strategy yielded 493 records from five common databases (MEDLINE via Ovid, Scopus, Web of Science, Psych Info, and ProQuest via ASSIA) and 4 additional papers were retrieved from other additional sources (Figure 1). Ten duplicates were removed electronically followed by a manual removal of nine more duplicates. Following records screening, 53 records were further excluded (Books (n = 34) and dissertations (n = 19)). Two reviewers (AA) and (GA) manually screened the title of the remaining 425 records and eliminated 394 records. A reliability test was carried out to evaluate the level of agreement for title screening and showed substantial agreement (K = 0.6). Then, the two reviewers screened the abstracts of the remaining 31 studies and further eliminated 14 records. The level of agreement was tested for abstract screening and revealed perfect agreement (K = 0.9). Thus, 17 full-text records were assessed for eligibility. Three records were excluded for not meeting the required age group [38,62,63]. The final review included 14 papers that fulfilled the inclusion criteria (Table 1). All studies were published from 2000 onwards. This review process was implemented separately and then results were compared and agreement was concluded. The third and the fourth reviewer (SC, EG) resolved discrepancies. Table 2 summarises the key information from the included studies. 

### 3.2. Quality Appraisal 

No studies were excluded based on quality assessment. Criteria for inclusion/exclusion (Q1) was clear for most of the studies, except for four studies [64,65,66,67], where no information was included on participants’ exclusion. Most studies met the criteria for the subjects and setting described in detail (Q2), excluding Al-Subaie [68], which was rated unclear. All studies met the third criterion (Q3) of measuring the exposure in a valid and reliable way, as they all applied validated self-reported questionnaires. The fourth criterion (Q4) concerned with objective and standard criteria used for measurement of the condition was met in the selected studies, except for Allihaibi [66], which was not clear. The seventh and eighth criteria were related to outcome measurement (Q7) and appropriate statistical analyses (Q8), and all 14 studies met their criteria [64,65,66,67,68,69,70,71,72,73,74,75,76,77]. Overall, all studies were classified as high quality, except for Allihaibi [66], which was classified as medium quality (Appendix A).

### 3.3. Study Characteristics

Studies involved students from schools and universities from distinct regions in Saudi Arabia, including central [68,74,77], western [65,66,69,70,75,76], eastern [72,73], southwestern [67], northwestern [64], and northern regions [71]. Variances in age and sex distribution were noted. Participants’ ages ranged from 10 to 24 years [78], with five studies examining university and college undergraduates, including two on females [73,74], two on both males and females [65,77], and one on males [72]. Nine records assessed school adolescent students, including eight on females [64,66,67,68,69,71,75,76], and only one on both sexes [70].

Most studies were focussed on female students [64,66,67,68,69,71,73,74,75,76]. Only three studies investigated both male and female students [65,70,77]; one record examined males [72]. Studies selected in this review identified the prevalence of different disorders categorised under EDs and DEB, distributed according to questionnaires implemented (EAT-26 [64,65,66,67,71,73,74,75,76,77], EDI-DT [68], EDE-Q [72], and SCOFF [70]), region, sex, and age.

### 3.4. Questionnaires

#### 3.4.1. Prevalence of Disorders Assessed by the EAT-26

The estimated prevalence of EDs/DEB for 11 studies was measured using EAT-26 [64,65,66,67,69,71,73,74,75,76,77]. Across studies, EAT-26 was the predominant tool used for evaluating aberrant eating behaviours. Of the 11 studies, 5 studies focussed on assessing the prevalence of EDs [65,67,73,76,77] and the other 6 determined the frequency of DEBs [64,66,69,71,74,75]. Among studies, the prevalence of these disorders was classified to be particularly high, ranging from 10.2% to 48.1%. As shown in Table 2, most studies included female students and college undergraduates. Only two studies examined both male and female students [65,77]. Differences between males and females were inconsistent with one study reporting a higher prevalence in females (38.0% females, 19.3% males [65]) and one in males (33.3% females, 40.0% males [77]).

#### 3.4.2. Prevalence of Disorders Assessed by the EDI-DT 

As shown in Table 2, only one study assessed the prevalence of dieting using the EDI-DT [68]. According to Al-Subaie [68], the study recruited 1179 female students aged 12 to 21 years (M = 16.13 years, SD = 2.09) who completed the EDI-DT. In proportion to students who scored more than 14, about 16% of the sample reported DEB.

**Table 2 nutrients-15-04643-t002:** Characteristics of studies included in this systematic review.

Author (Year)	City/ Area	Study Design	Setting	Sample Size	Sex Distribution (F/M)	Age Group (Mean Age ± SD)	Screening Tool	Disorder Screened	Prevalence (N and/or %)	Quality Assessment Score
Albrahim et al. (2019) [74]	Riyadh/central	Cross- sectional	Undergraduate students from Prince Nora University	396	396/0	18–24 (20.1 ± 1.55)	EAT-26	DEAs	EAT-26: ≥20 (n = 145, 36.8%), impulse to vomit after meals (2.08 ± 1.14), dieting (0.26 ± 0.66), binging (0.52 ± 0.92)	High
AlHazmi and AlJohani (2019) [65]	Madina/west	Cross-sectional	Health specialties students at Taiba University	342	171/171	NR (NR) ≤22 (n = 221, 64.4%)	EAT-26	EDs	EAT-26: ≥20 (n = 75, 33.9%), F (n = 65, 38%), M (n = 33, 19.3%)	High
Allihaibi (2015) [66]	Makkah/west	Cross-sectional	Secondary schools	180	180/0	15–19 (16.83 ± 0.94)	EAT-26	DEAs	EAT-26: ≥20 (n = 47, 26.1%)	Medium
Almuhlafi et al. (2018) [64]	Tabuk/northwest	Cross-sectional	High schools	399	399/0	NR (16.8 ± 0.9)	EAT-26	DEBs: Binge eating, purging, laxatives	EAT-26: ≥20 (n = 192, 48.1%) DEB: Binge eating (n = 123, 30.8%), purging (n = 28, 7%), laxative use (n = 21, 5.3%)	High
Al-Qahtani and Al-Harbi (2020) [69]	Madina/west	Cross-sectional	Secondary schools	393	393/0	15–20 (17.24 ± 1.03)	EAT-26	DEBs	EAT-26: ≥20 (n = 167, 42.5%), self-reported frequency of EA in the last 6 months (binge eating: (≤3 = 43.4%, >3 = 56.6%) SIV: (≤3 = 44.1%, >3 = 55.9%), laxatives or diuretics: (≤3 = 45.8%, >3 = 54.2%)	High
Alsubaie et al. (2017) [67]	Abha/southwest	Cross-sectional	Intermediate and secondary schools	224	224/0	12–19 (15.9 ± 3.7)	EAT-26	EDs	EAT-26: ≥20 (n = 85, 34%), older age females (17–19) had a significantly higher score of EAT-26 (n = 35, 43.8%) than the younger age (12–16) group (n = 50, 29.4%)	High
Al-Subaie (2000) [68]	Riyadh/central	Cross-sectional	Grades 7 to 11 from Intermediate and Secondary schools	1179	1179/0	12–21 (16.13 ± 2.09)	EDI-DT	DEB: Dieting	EDI-DT > 14 (n = 188, 15.9%)	High
Alwosaifer et al. (2018) [73]	Dammam/East	Cross-sectional	Imam Abdulrahman bin Faisal university	656	656/0	18–23 (18.7 ± 0.74)	EAT-26	EDs	EAT-26: ≥20 (n = 179, 29.4%), problematic eating behaviours (n = 277, 45.5%)	High
ElShikieri (2022) [75]	Madina/west	Cross-sectional	Public and private female elementary, intermediate, and high schools	381	381/0	10–18 (M = 13.6, SD = 2.6)	EAT-26	DEAs	EAT-26: ≥20 (n = 39, 10.2%)	High
Fallatah et al. (2015) [76]	Jeddah/west	Cross-sectional	Secondary schools	425	425/0	15–18 (16.6 ± 0.98)	EAT-26	EDs	EAT-26: ≥20 (n = 140, 32.9%)	High
Fatima and Ahmed (2018) [71]	Arar/north	Cross-sectional	Schools	314	314/0	15–19 (17.0 ± 1.14)	EAT-26	DEAs	EAT-26: ≥20 (n = 80, 25.4%)	High
Ghafouri et al. (2021) [70]	Makkah/west	Cross-sectional	Private and public secondary schools	471	399/72	NR (17.28 ± 1.27)	SCOFF	EDs	SCOFF ≥ 2 (n = 136, 46%), SCOFF 0 or 1 (n = 216, 29%)	High
Loni et al. (2022) [77]	Majmaah/central	Cross-sectional	Majmaah University	125	90/35	18–25 (F (22.9 ± 4.9)), M (25.6 ± 5.1)	EAT-26	EDs	EAT-26: ≥20 (n = 44, 35.2%), M (n = 14, 40%), F (n = 30, 33.3%), binge eating (M = 42.9%, F = 57.8%), exercised more than 60 min (M = 60%, F = 54.4%), SIV (M = 5.7%, F = 13.3%)	High
Tomar and Antony (2022) [72]	Dhahran/East	Cross-sectional	King Fahad University for petroleum and minerals	60 (obese participants (BMI) ≥ 30 kg/m^2^)	0/60	18–25 (19.67 ± 0.90)	EDE-Q	EDs	High EDE-Q global score (n = 36, 65.5%)	High

NR: Not reported, EDs: Eating Disorders, DEAs: Disordered Eating Attitudes, DEBs: Disordered Eating Behaviours, EAT-26: Eating Attitudes Test, EDE-Q: Eating Disorder Examination- Questionnaire, EDI-DT: Eating Disorder Inventory Drive for Thinness, SCOFF: (Sick, Control, One, Fat, Food) questionnaire, SIV: Self-induced Vomiting.

#### 3.4.3. Prevalence of Disorders Assessed by the EDE-Q

Only one study investigated the prevalence of EDs using EDE-Q [72]. Tomar and Antony [72] recruited 60 university students with obesity in Dhahran (eastern province in Saudi Arabia) aged 18 to 25 years. EDE-Q scores indicated that 36 out of 60 participants (65.5%) are considered to have a high prevalence of EDs.

#### 3.4.4. Prevalence of Disorders Assessed by SCOFF

As demonstrated in Table 2, only one study implemented the SCOFF questionnaire on private and public secondary school students [70]. Adolescent boys (n = 72) and girls (n = 399) were selected in Makkah (western province in Saudi Arabia) to examine the prevalence of EDs by applying the SCOFF questionnaire. According to the SCOFF scores, Ghafouri and colleagues classified that almost half of the sample (46.0%) scored two or more and might be anorexic or bulimic [70].

#### 3.4.5. Region

As mentioned in Table 2, six studies were conducted in the western region. Specifically, three studies were implemented in Madina [65,69,75], two studies in Makkah [66,70], and one in Jeddah [76]. Three studies were administered in the central region, including two studies in Riyadh [68,74] and one in Majmaah [77]. In the eastern region, only two studies were implemented, one in Dammam [73] and one in Dhahran [72]. One study was found in the southwestern region, in the city of Abha [67]. Similarly, one study was implemented in the northwestern region, in Tabuk [64], and one study was administered in the northern region, in Arar [71]. The prevalence of EDs/DEB in the western region is classified as ranging from 10.2% to 46.0% [65,66,69,70,75,76]. According to the central region, the prevalence ranged from 15.9% to 36.8% [68,74,77]. The prevalence in the eastern region was reported to range from 29.4% to 65.5% [72,73]. In the southwestern, northwestern, and northern regions, the prevalence was reported as 34.0% [67], 48.1% [64], and 25.4% [71], respectively.

#### 3.4.6. Sex

Ten studies were administered on female students [64,66,67,68,69,71,73,74,75,76]. Three studies recruited both females and males [65,70,77] and one study was implemented on male students [72]. The prevalence of disorders ranged from 10.2% to 48.1% among female students [64,65,66,67,68,69,70,71,73,74,75,76,77] and from 19.3% to 65.5% among male students [65,70,72,77]. 

#### 3.4.7. Age

As clarified in Table 2, the study setting involved school and university students. Nine studies recruited school students with ages ranging from 10 to 21 years [64,66,67,68,69,70,71,75,76]. Five studies recruited university students aged from 18 to 25 years [65,72,73,74,77]. For school students, the reported prevalence of ED/DEB was from 10.2% to 48.1% compared to university students who reported a prevalence ranging from 29.4% to 65.5%. 

## 4. Discussion

This is the first systematic review on the prevalence of EDs and DEB among students and young adults in Saudi Arabia from 2000 onwards. This review was implemented to explore the epidemiology of EDs and DEB among adolescents and young adults in Saudi Arabia with the aim of assessing the prevalence of EDs/DEB in Saudi adolescents and young adults according to the type of questionnaires, region, sex, and age. 

The prevalence of EDs/DEB seems to be a major concern in Saudi Arabia. In this review, different regions of Saudi Arabia were investigated to assess the prevalence of EDs/DEB according to the questionnaires applied (EAT-26, EDI-DT, EDE-Q, and SCOFF). About 79% of studies (11 out of 14) utilised EAT-26 to determine the presence of such disorders, with a prevalence ranging from 10.2% to 48.1% [64,65,66,67,69,71,73,74,75,76,77]. The other tools include the EDI-DT, SCOFF, and EDE-Q, yielding prevalence rates of 15.9%, 46.0%, and 65.5%, respectively [68,70,72]. Amongst studies, EAT-26 was the dominantly used questionnaire to identify the prevalence of DEB. It is evident that different questionnaires produced different prevalence estimates between studies, which could be linked to the age, sex, or social/cultural/economic/regional factors of the studied population [64,65,66,67,68,69,70,71,72,73,74,75,76,77]. Differences can be also attributed to study methods such as the recruitment process and administration of questionnaires [64,65,66,67,68,69,70,71,72,73,74,75,76,77]. Comparing types of questionnaires, SCOFF showed no significant difference in detecting prevalence rates when compared to EAT-26, and both questionnaires’ scores were positively associated [79]. Another study compared EAT, EDI, and EDE-Q questionnaires among female adolescents, confirmed that they perform well in estimating prevalence, and also indicated that EDI is the questionnaire with the highest sensitivity and specificity [80].

Equivalent proportions were identified from other countries in the Middle East. Depending on EAT-26/EAT-40, students from UAE reported a prevalence of DEB from 23.4% to 41.2% [43,81,82]. Moreover, using EAT-26 as a screening tool for DEBs, it has been reported that the prevalence of such disorders among high school students was 31.6%, 44.7%, 26.7%, 31.7%, and 22.9% in Jordan, Kuwait, Libya, Palestine, and Syria, respectively [43]. Another study on Jordanian students found that 40.5% reported DEB depending on EAT-26 scores [83]. Al-Adawi et al. concluded that 29% of Omani school students reported DEB [84]. References from Iran indicated that DEB prevalence ranged from 15% to 24.2% [85,86,87,88]. Surprisingly, Egypt indicated the highest proportion of DEB, with 73.3% reported prevalence [89]. Considering Egypt as a less economically developed country scoring higher rates than other developed countries, the author explained that overreporting in self-administered questionnaires is a possibility that could not be avoided [89]. Overall, EAT-26 findings from countries in the Middle East and Saudi Arabia were comparable. Additionally, research from the United States, Europe, and Asia is frequently comparable to findings from Saudi Arabia [20,25,36,37].

### 4.1. Region

Most studies were conducted in the western region of Saudi Arabia including Madina [65,69,75], Makkah [66,70], and Jeddah [76], with a reported DEB prevalence ranging from 10.2% to 46%. EAT-26 was the predominant questionnaire applied in western region studies, except for Ghafouri et al., who conducted the SCOFF questionnaire on secondary school students and found the highest prevalence of EDs in the region (46%) [70]. There is no clear explanation for the western region’s predominance in studies, which could be due to feasibility and convenience for data collection.

Three studies were implemented in the central region [68,74,77], followed by two studies in the eastern region [72,73]. One study was carried out in the southwestern region [67], one in the northwestern region [64], and one in the northern region [71]. Although a meta-analysis was not conducted, it was still evident from the individual study results that the overall prevalence was highest in the eastern region ranging from 29.4% to 65.5%, followed by the northwestern region (48.1%), and the lowest prevalence was found in the northern regions (25.4%).

Regions in Saudi Arabia reported comparable rates of EDs/DEB with the eastern region reporting the highest prevalence. Despite the differences in selected cities in relation to size, climate, and population count, they all demonstrated a high-risk prevalence of EDs/DEB. This is in contrast to Takimoto et al., which found levels of thinness among adolescent girls aged 15–19 years to be higher in larger cities compared to smaller cities [90]. They also indicated that adolescents living in larger cities have more possibility of engaging in DEB compared to their peers who are residing in smaller cities [90]. In relation to urbanisation and affluence, there was a lack of variation between studied cities in relation to prevalence rates. The central region has the highest levels of urbanisation and affluence [68,74,77], followed by the eastern and western regions, although this was not projected on prevalence rates [65,66,69,70,72,73,75,76]. These conclusions were unexpected and worth further investigation. A recent scoping review on urbanisation and EDs demonstrated that urbanisation is not directly associated with an increased prevalence of such disorders [49].

### 4.2. Sex

In relation to sex, the majority of studies recruited female students [64,66,67,68,69,71,73,74,75,76], with only four studies including male students [65,70,72,77]. Tomar and Antony focussed their research on male university students only [72], while the other three records selected both female and male students [65,70,77]. Figures on males indicate that DEB is apparent, with AlHazmi and AlJohani reporting that 19.3% reported DEB and Tomar and Antony had results as high as 65.5% [65,72]. Ghafouri et al. interpreted results for the total sample and did not report them according to sex [70]. Surprisingly, Loni et al. confirmed that males reported a higher prevalence of DEB (40%) compared to female participants (33.3%), which is contradictory to many findings that classify females with higher prevalence [77].

Overall, the focus on females in most studies is expected, as the literature continuously stresses on females being more concerned with their body weight and exposed to a higher prevalence of DEB [20,24,25,39,81,91,92]. 

Females are more likely to have disordered consumption, and the literature has attention on females globally [39,91,92]. A study assessing the prevalence of EDs among medical students in Karachi, Pakistan found that 87.9% of females and 12.1% of males were classified as being at high risk of developing an ED [93]. Another study examined the prevalence of DEB in a considerable European sample and concluded that females tend to be 2.96 to 5.9 times more prone to practicing DEB than males [94]. In Finland, results confirmed that young females have higher odds compared to young males to be affected by EDs [95]. Studies in Iran, Jordan, Palestine, and Israel have concluded similar findings [40,87,96,97,98,99,100]. In this review, the results were not conclusive as Loni et al.’s conclusions were contradictory [77]. However, it is essential to address that EDs/DEB is common among Saudi male adolescents and young adults and might be underreported [62,65,70,72,77]. Young males with EDs/DEB explained that these feminised cultural constructs of EDs inclined them (as well as those around them—relatives, peers, and health and education experts) to ignore symptoms [101].

### 4.3. Age

In terms of age, this review indicated that the reported prevalence ranged from 29.4% to 65.5% for university students and 10.2% to 42.5% for secondary and intermediate school students [64,65,66,67,69,71,73,74,75,76,77]. Clearly, older adolescents have higher prevalence of EDs/DEB compared to younger adolescents. Alsubaie et al. clarified that Saudi older female students scored higher prevalence rates of DEB (43.8%) compared to younger students (29.4%) [67]. These findings support conclusions that indicate that the peak of DEB occurs in late adolescence and early adulthood [33].

University/college students encountered a higher prevalence of EDs/DEB when compared to school students. A study on the UK population indicated that the peak age for developing disordered behaviours ranges from 15 to 19 years [102]. Another study investigated the estimated prevalence of EDs in a US cohort and found that the prevalence of these disorders was reported to be the highest at the age of 21 years for both sexes [33]. The majority of first-time cases (95%) are developed by the age of 25 years [33]. Inconsistent findings were reported among Korean adolescents, as they concluded that DEB and distorted body image were more prevalent in adolescents aged 10–17 years than young adults (18–24 years), with the highest prevalence found among females aged 10–12 years [103]. A clear contradiction was found in relation to age and EDs, as some studies clarified such behaviours to be common among younger adolescents, while other studies confirmed they are most prevalent in older adolescents [104].

Based on our review, Saudi Arabia’s prevalence estimates tend to be also higher than those reported in other Asian and European countries [105,106,107,108,109,110,111,112,113,114]. Several studies on Turkish school and university students revealed that the prevalence of DEB ranged from 9.8% to 22.8% [105,106,107,108]. In Japan, the Japanese version of EAT-26 confirmed that 11.2% of females experienced eating problems [109]. In Korea and Taiwan, disturbed eating attitudes were recorded in 10.3% and 10.4% of students, respectively [110,111]. Szweda and Thorne confirmed a marked prevalence of EDs amongst all female student groups in the UK [112]. They revealed that 20% of nursing students showed DEB patterns with no significance reported when compared to medical students (19%) and art students (21%). In the US, up to 26% of school students had elevated EAT-26 scores [113]. In Canada, 16% of females aged 15–18 years and 13% of females aged 12–14 years scored above the cut-off point in the EAT-26 [114]. Collectively, rates of EDs/DEB from Asian countries, the UK, the US, and Canada appear to be lower than indicated in Saudi Arabia [112,113,114].

Originally, research indicated that DEBs and related disorders have been acknowledged in Western countries since the late 20th century [56,115]. Early studies claimed that an ED was a common diagnosis in white women from Western countries with only a few cases recognised from non-Western countries [116]. However, recent studies demonstrated that DEB problems are increasing dramatically in non-Western countries, particularly in Saudi Arabia [38,63,64,65,69,76,116]. There is no clear rationale to explain the marked rise in ED/DEB conditions in Saudi Arabia, as it relies on complex etiologic interactions within genetic, social, cultural, and psychological causes that elevate the possibility of encountering these distinct, culturally bound syndromes [117].

Many researchers delineated extended trends of EDs/DEB to be affected by Westernisation and the preference of a thin body physique [87,117]. Culturally, Arab populations including Saudi Arabia valued plumpness, as they believed it conveys fertility, wealth, and health [118,119]. However, Saudi Arabia and other Middle Eastern countries have experienced dramatic economic developments and sociocultural shifts in the last decades [71,120,121]. Consequently, Saudi adolescents and young adults are exchanging Arabic traditional values and adopting Western concepts in relation to weight and physical appearance, which can negatively influence the possibility of developing EDs/DEB in young generations [39,48,81].

### 4.4. Strengths and Limitations

This is the first review to assess the prevalence of EDs/DEB among Saudi adolescents and young adults. Records selected for this review revealed consistency in terms of study design, setting, and screening tool [64,65,66,67,68,69,70,71,72,73,74,75,76,77].

Self-reported questionnaires were used as screening tools to evaluate the prevalence of such disordered behaviours. Questionnaires are practical tools but appear to be insufficient in confirming an ED diagnosis, and suspected cases require further clinical assessment [122]. Thus, only studies that applied self-reported instruments were selected. Other methods might also be practical in examining the presence of DEB, including interviews and weight-history taking.

Although some studies included the target age group (10–24 years) identified in the inclusion/exclusion criteria, they had to be excluded in the selection process, since they decided that overall conclusions were connected with older participants’ findings (faculty members and employees), limiting the ability to draw inferences on adolescents and young adults [38,62,63]. Most selected studies focussed on female adolescents, as they are more vulnerable to engaging in DEB [20,24,25]. Particularly, four of the selected studies included male adolescents and yielded contradictory results [65,70,72,77].

In terms of region, most studies were conducted in the western region of Saudi Arabia, which could limit the generalisability of the findings to the wider Saudi population [65,66,69,70,75,76].

Despite how each study presented valuable findings in relation to DEB among school students and adolescents in Saudi Arabia, the cross-sectional nature of the studies increases the possibility of bias and may limit the generalisability of the findings [123]. Further research on adolescent males and young people is essential to fill the gaps in the present epidemiological evidence and develop a thorough contextual comprehension of the nature of ED/DEB on the whole population.

In relation to the inclusion/exclusion criteria, some studies did not exclude participants with chronic physical and mental conditions [64,65,66,67]. Yet, they did not plan any considerations for them. A specific ED/DEB could be the result of another ailment. It is necessary to exclude them and analyse the prevalence in those with another condition separately [124,125,126].

These studies were all characterised to have a one-stage process, particularly in relation to the self-reported questionnaires [64,65,66,67,68,69,70,71,72,73,74,75,76,77]. No clinical interviews or diagnoses were applied. Few studies linked the lack of clinical diagnostic interventions with limited resources [66,71].

In some studies, the limited sample size may pose difficulty in generalising the results to general populations [64,66,67,71,75]. Self-reported questionnaires are considered applicable and affordable tools; however, they have a likelihood of overestimation and may be prone to bias [127]. Moreover, it is difficult to exclude to possibility of publication bias, for example, it may be possible studies finding a high prevalence in males are more likely to be published.

## 5. Conclusions

To our knowledge, this systematic review is the first to explore the prevalence of EDs/DEB among adolescents and young adults in Saudi Arabia and provide a clear comprehension of the nature of EDs/DEB by describing them among the Saudi population. These disorders are found to be a significant problem in this country and the risk of developing such conditions is high across the whole age range included in this review. Therefore, this noticeable increase in Saudi Arabia must motivate researchers, physicians, and dietitians to develop scientifically based guidelines for assessing, evaluating, and treating these unhealthy behaviours. Saudi adolescents and young adults require educational awareness of healthy food consumption and the possible consequences of DEB.

## Figures and Tables

**Figure 1 nutrients-15-04643-f001:**
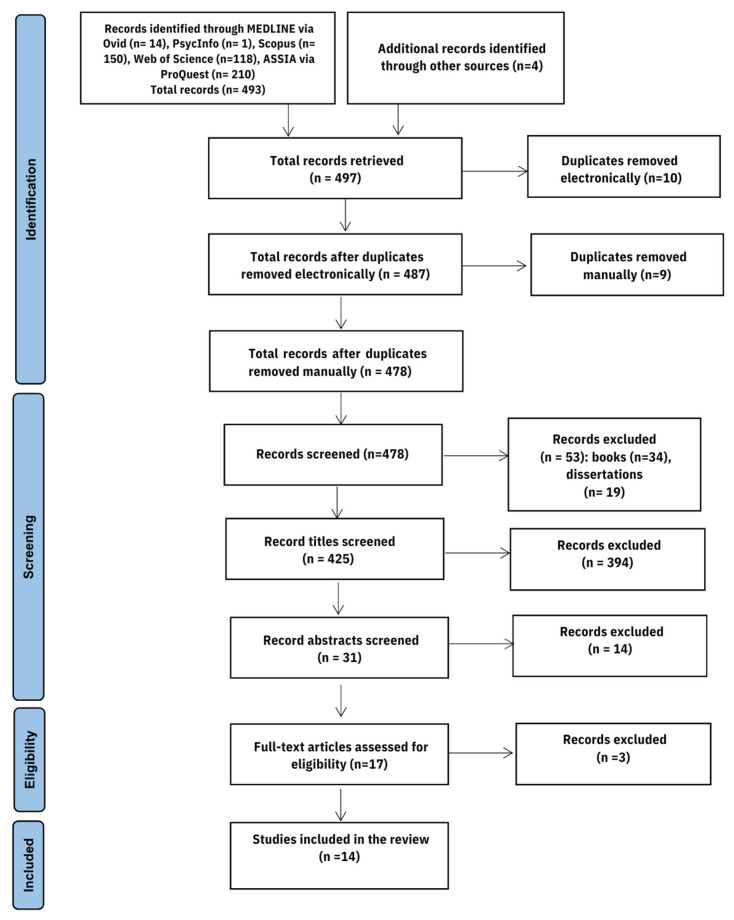
PRISMA flow diagram showing the process of study selection [50].

**Table 1 nutrients-15-04643-t001:** Inclusion and exclusion criteria for studies included in the review.

Components	Inclusion Criteria	Exclusion Criteria
Population	Healthy adolescent and young adult participants aged 10–24 years.	Participants with certain metabolic or mental/psychiatric disorders (Attention Deficit Hyperactivity Disorder (ADHD), Obsessive Compulsory Disorder (OCD)).
Exposure	Studies assessing eating disorders, disordered eating, feeding disorders, appetite disorders, anorexia nervosa or bulimia nervosa, binge eating, disturbed eating, weight control, or purging.	Studies assessing orthorexia disorder, Pica, and rumination disorder.
Context	Collected samples from Saudi Arabia	Samples from outside Saudi Arabian borders
Outcome	Studies measuring the prevalence of EDs/DEB with a validated tool: self-reported questionnaire (EAT-26, EDE-Q…)	Studies focussed on clinical assessment, interviews, or weight-history taking to assess the prevalence of ED/DEB
Study design	Cross-sectional and longitudinal studies are both eligible. Journal articles published in peer-reviewed journals and conference papers.	Qualitative studies (case reports), books, editorials, dissertations, systematic/narrative reviews. Studies published in Arabic with no available English translation since English is regarded as the universal language of science.

## Data Availability

Not applicable.

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
