# Peer review of "Prevalence of Eating Disorders and Disordered Eating Behaviours amongst Adolescents and Young Adults in Saudi Arabia: A Systematic Review"

_nutrients, 2023, doi:10.3390/nu15214643_

Round 1

Reviewer 1 Report

Comments and Suggestions for Authors

I have carefully reviewed this manuscript with interest, and the results are as follows.

1.      ‘Disruptive Eating Behaviors (DEB)’ described in the research topic and ‘disordered eating behaviors (DEB)’ presented in the abstract and body of the manuscript are not the same concept. In addition to confirmation of this, consistent application is required. Besides, a theoretical basis must be presented that Eating Disorders (ED) can be included in Disruptive Eating Behaviors (DEB).

2.      Clear theoretical and operational definitions of the terms applied in this study are needed. Those terms are: DE: Disordered Eating , DEA: Disordered Eating Attitudes, DEB: Disordered Eating Behaviors, EA: Eating Attitudes, ED: Eating Disorders etc.

3.      In materials and methods, there are 14 previous studies included in the SR, and the theoretical basis for conducting the SR must be presented in detail. In addition, the reason why meta-analysis was not performed should be described.

4.      When performing this SR, it must be presented whether it was implemented according to PICOTS-SD.

5.      Describe the authors' SR performance competency.

6.      The reason for conducting the data synthesis on ED and DEH should be described, and it is reasonable to integrate the previous papers separately.

7.      Reasons for not conducting the piolt test for literature selection and exclusion must be presented.

8.      The rationale for selecting variables such as sex, age, and region for subgroup analysis should be clearly presented, and the year of publication should be added.

9.      The basis for reporting the highest prevalence of ED/DEB in the eastern region of Saudi Arabia, without performing a meta-analysis, should be described.

10.   In this SR, the research finding was derived by applying the level of education instead of the age variable. Calculate the value of the dependent variable according to the age group. Also, explain the reason for calculating the result by replacing age with the level of education.

11.   The publication bias of previous studies should be described.

12.   In Table 1, the rationale for the components selected as inclusion criteria should be presented.

13.   This study concludes that disorders such as ED/DEB are more prevalent in Saudi Arabia than in other countries and that the prevalence of these conditions is comparable to other countries. This conclusion is possible through a meta-analysis of previous studies conducted not only in Saudi Arabia but also in other countries. A correction is needed for this.

14.   The description of the preceding research cited in the text does not conform to the format of this journal, so please revise it. For example, Neumark-Sztainer et al. [20] explained that~.

15.   The reference list does not conform to the format of this journal, so please edit all of them.

Author Response

Dear reviewer,

Thank you for sending your comments. Please see the attached file below.

Response Letter

Manuscript ID nutrients-2553218

Title Prevalence of Disruptive Eating Behaviours Amongst Adolescents and Young Adults in Saudi Arabia: A Systematic Review

Reviewer 1

Thank you for your comments. The table below contains amendments that address the comments provided and the uploaded revised manuscript reflects the revisions made according to the reviewers' remarks.

Comment

Amendment

 ‘Disruptive Eating Behaviors (DEB)’ described in the research topic and ‘disordered eating behaviors (DEB)’ presented in the abstract and body of the manuscript are not the same concept. In addition to confirmation of this, consistent application is required. Besides, a theoretical basis must be presented that Eating Disorders (ED) can be included in Disruptive Eating Behaviors (DEB).

To be consistent and avoid using terms that reduce the level of clarity, the definition (Disruptive Eating Behaviours) was removed from the research topic and modified to ‘’ Prevalence of Eating Disorders (ED) and Disordered Eating Behaviours (DEB) Amongst Adolescents and Young Adults in Saudi Arabia: A Systematic Review’’. The term disruptive eating was also omitted throughout the paper, and the terms (Eating disorders ED) and (Disordered eating behaviours DEB) were only used.

 Clear theoretical and operational definitions of the terms applied in this study are needed. Those terms are: DE: Disordered Eating , DEA: Disordered Eating Attitudes, DEB: Disordered Eating Behaviors, EA: Eating Attitudes, ED: Eating Disorders etc.

Terms were modified to (Eating disorders ED) and (Disordered Eating Behaviours DEB). All other similar terms were removed. The terms ED and DEB were fully explained.

In the table, the disorders screened were identified according to the selected papers’ purpose which included ED, DEB, and DEA. However, DEB and DEA included the same behaviours.

In materials and methods, there are 14 previous studies included in the SR, and the theoretical basis for conducting the SR must be presented in detail. In addition, the reason why meta-analysis was not performed should be described.

3. Results

3.1. Study Selection

The initial search strategy yielded 493 records from five common databases MEDLINE via Ovid, Scopus, Web of Science, Psych Info, and ProQuest via ASSIA, and four additional papers were retrieved from other additional sources (Figure 1). Ten duplicates were removed electronically followed by a manual removal of nine more duplicates. Following records screening, 53 records were further excluded (Books (n=34) and dissertations (n= 19)). Two reviewers (AA) and (GA) manually screened the title of the remaining 425 records and eliminated 394 records. Then, the two reviewers screened the abstracts of the remaining 31 studies and further eliminated 14 records. Thus, 17 full-text records were assessed for eligibility. Three records were excluded for not meeting the required age group [38,62,63]. The final review included 14 papers that fulfilled the inclusion criteria (Table.1). All studies were published from 2000 onwards.  This review process was implemented separately and then results were compared and agreement was concluded. The third and the fourth reviewer (SC, EG) resolved discrepancies. Table 2 summarizes the key information from the included studies.

2.2 Inclusion and Exclusion Criteria

As the inclusion criteria resulted in the identification of studies using a range of different methods to measure prevalence, the heterogeneity of studies meant that meta-analysis was not considered appropriate.

.      When performing this SR, it must be presented whether it was implemented according to PICOTS-SD.

1.2  Inclusion and Exclusion Criteria

Studies were assessed and included based on the following eligibility criteria following the PICOs from the JBI manual for evidence synthesis [51] (Table.1).

Describe the authors' SR performance competency.

2.3 Screening and Data Extraction

The reason for conducting the data synthesis on ED and DEH should be described, and it is reasonable to integrate the previous papers separately.

1.       Introduction

No previous review has focused on such disorders in populations in Saudi Arabia. In this regard, a systematic review is imperative to develop a clear understanding of the epidemiology of ED and DEB among adolescents and young adults in Saudi Arabia.

Reasons for not conducting the pilot test for literature selection and exclusion must be presented.

3. Results

3.1. Study Selection

A reliability test was carried out to evaluate the level of agreement for title screening and showed substantial agreement (K= 0.6).

The level of agreement was tested for abstract screening and revealed perfect agreement (K=0.9)

The rationale for selecting variables such as sex, age, and region for subgroup analysis should be clearly presented, and the year of publication should be added.

1.Introduction

As this is an exploratory study, these variables (sex, age, region) were emphasized to understand the demographic differences in relation to the prevalence of ED/DEB in Saudi Arabia.

As suggested, the year of publication is added for each record in the table

The basis for reporting the highest prevalence of ED/DEB in the eastern region of Saudi Arabia, without performing a meta-analysis, should be described.

As suggested, a statement was added to explain the highest prevalence reported without implementing a meta-analysis.

4. Discussion

Region

Although a meta-analysis was not conducted, it was still evident from the individual study results that the overall prevalence was highest in the eastern region ranging from 29.4% to 65.5%, followed by the northwestern region (48.1%), and the lowest prevalence was found in the northern regions (25.4%).

In this SR, the research finding was derived by applying the level of education instead of the age variable. Calculate the value of the dependent variable according to the age group. Also, explain the reason for calculating the result by replacing age with the level of education.

We appreciate that educational level is used as a proxy for age as well as describing the setting for the included studies and we have used it as we were particularly interested in identifying whether prevalence was different in school and college/university settings. As suggested, this was clarified in the section below.

2.5 Data Synthesis

It is important to note that educational setting is used rather than age group per se because the main interest was identifying whether prevalence was different in school and college/university settings.

The publication bias of previous studies should be described.

As suggested, we have addressed the problem of potential publication bias in the section below

4.3 Strengths and Limitations

Moreover, it is difficult to exclude to possibility of publication bias, for example, it may be possible studies finding a high prevalence in males are more likely to be published.

 In Table 1, the rationale for the components selected as inclusion criteria should be presented.

As suggested, the rationale is mentioned in the section below.

2.2 Inclusion and Exclusion Criteria

As clarified in the table, only studies that serve the aim of this systematic review are included, recruiting healthy adolescents from Saudi Arabia and defining the prevalence of ED/DEB using a validated tool.

This study concludes that disorders such as ED/DEB are more prevalent in Saudi Arabia than in other countries and that the prevalence of these conditions is comparable to other countries. This conclusion is possible through a meta-analysis of previous studies conducted not only in Saudi Arabia but also in other countries. A correction is needed for this.

As suggested, the statement is modified to ‘’These disorders are found to be a significant problem in this country and the risk of developing such conditions is high across the whole age range included in this review .’’

The description of the preceding research cited in the text does not conform to the format of this journal, so please revise it. For example, Neumark-Sztainer et al. [20] explained that~.

As suggested, the format has been corrected.

The reference list does not conform to the format of this journal, so please edit all of them.

As suggested, references have been edited to ACS referencing style.

Kind regards,

Azzah Alsheweir

Reviewer 2 Report

Comments and Suggestions for Authors

The work is interesting, well-conducted and written.

I advise authors to check and correct the following in the text of their scientific paper:

In the text contained in the pdf file of the paper, there seems to be a double space in the following lines: 18, 19, 145, 279, 353, 459, 534.

In the text contained in the pdf file of the paper, bibliographical references are sometimes written in [x, x] format, sometimes in [x,x] format. I ask the authors to align all text with the form [x,x].

Line 59: striction [11, 13] = Add a space

Line 61, 75: The paper sometimes refers to DSM-IV, sometimes to DSM-V, and sometimes only DSM. I ask authors to check this and specify IV or V when writing DSM.  

Line 82: used. = Remove a space

Line 88: ance [24] = Remove the first point

Table 1: In the text, the sentences all begin with the capital letter, except: participants (you use P), cross-sectional (you use C)

Line 175: Age+SD (remove a space)

Figure 1: The image, by printing the pdf, is blurred. It is recommended to improve the image quality (higher resolution).

Table 2: I would write EAT-26: >20 in the prevalence column. I think you want to indicate when that test scored above 20 points. Is it true?

Line 307, 309: [..77] and = Add a space

Line 311, 312: You must remove two entries to make the write format equal to the previous ones.

Line 384: 15-19 = Remove a space

Line 465: 20th = th in apice

Author Response

Dear reviewer,

Thank you for sending your comments. Please see the attached file below.

Response Letter

Manuscript ID nutrients-2553218

Title Prevalence of Disruptive Eating Behaviours Amongst Adolescents and Young Adults in Saudi Arabia: A Systematic Review

Reviewer 2

Thank you for your comments. The table below contains amendments that address the comments provided and the uploaded revised manuscript reflects the revisions made according to the reviewers' remarks.

Comment

Amendment

In the text contained in the pdf file of the paper, there seems to be a double space in the following lines: 18, 19, 145, 279, 353, 459, 534.

As suggested, double spaces are corrected.

In the text contained in the pdf file of the paper, bibliographical references are sometimes written in [x, x] format, sometimes in [x,x] format. I ask the authors to align all text with the form [x,x].

As suggested, spaces are removed. The format [x,x] was applied to the whole paper.

Line 59: striction [11, 13] = Add a space

As suggested, a space is added.

Line 61, 75: The paper sometimes refers to DSM-IV, sometimes to DSM-V, and sometimes only DSM. I ask authors to check this and specify IV or V when writing DSM.  

As suggested, specifications for DSM are checked. In this paper, both DSM-IV and DSM-V are used to compare the differences between the old manual and the updated one, particularly in relation to binge eating and its diagnosis.

Line 82: used. = Remove a space

As suggested, the space is removed.

Line 88: ance [24] = Remove the first point

As suggested, the first point is removed.

Table 1: In the text, the sentences all begin with the capital letter, except: participants (you use P), cross-sectional (you use C)

As suggested, letters are modified to capital letters.

Line 175: Age+SD (remove a space)

As suggested, the space is removed.

Figure 1: The image, by printing the pdf, is blurred. It is recommended to improve the image quality (higher resolution).

As suggested, Figure 1 is modified.

Table 2: I would write EAT-26: >20 in the prevalence column. I think you want to indicate when that test scored above 20 points. Is it true?

As suggested, the format in Table 2 is corrected.

It indicates the prevalence of students who scored 20 or more on the EAT-26 test (high risk of ED/DEB).

Line 307, 309: [..77] and = Add a space

As suggested, a space is added.

Line 311, 312: You must remove two entries to make the write format equal to the previous ones.

As suggested, the two entries are removed.

Line 384: 15-19 = Remove a space

As suggested, the space is removed

Line 465: 20th = th in apice

As suggested, the format is corrected to 20th.

Kind regards,

Azzah Alsheweir

Reviewer 3 Report

Comments and Suggestions for Authors

I anonymously read the manuscript entitled “Prevalence of Disruptive Eating Behaviours Amongst Adolescents and Young Adults in Saudi Arabia: A Systematic Review” and I think that it is very interesting because the trated topic is current and it is represent a great concern for the health. However, before publication, some revisions are necessary to enhance the readability and scientific validity of the manuscript. Point-by-point comments are listed below:

The present language is not enough for publication in this journal and some sentences are difficult to understand. So, I suggest carefully revising the language throughout the manuscript.

The manuscript has too many keywords. Please write keywords according to the requirements of the journal.

I noticed that some conclusions are missing references, please check the manuscript and provide references accordingly.

The clarity of the Figure1 could be further improved.

Please revise the format of references according to the requirements of the journal.

Comments on the Quality of English Language

The present language is not enough for publication in this journal and some sentences are difficult to understand. So, I suggest carefully revising the language throughout the manuscript.

Author Response

Dear reviewer,

Thank you for sending your comments. Please see the attached file below.

Response Letter

Manuscript ID nutrients-2553218

Title Prevalence of Disruptive Eating Behaviours Amongst Adolescents and Young Adults in Saudi Arabia: A Systematic Review

Reviewer 3

Thank you for your comments. The table below contains amendments that address the comments provided and the uploaded revised manuscript reflects the revisions made according to the reviewers' remarks.

Comment

Amendment

The present language is not enough for publication in this journal and some sentences are difficult to understand. So, I suggest carefully revising the language throughout the manuscript.

As suggested, we have taken the opportunity to revise our manuscript for clarity and the paper has been edited by the Cambridge Proofreading and Editing team.

The manuscript has too many keywords. Please write keywords according to the requirements of the journal.

Nine keywords have been added to meet the required guidelines of the journal.

I noticed that some conclusions are missing references, please check the manuscript and provide references accordingly.

Additional references have been added as suggested.

The clarity of the Figure1 could be further improved.

As suggested, Figure 1 has been modified for clarity.

Please revise the format of references according to the requirements of the journal.

The format of references has been modified to the required ACS referencing style.

Kind regards,

Azzah Alsheweir

Round 2

Reviewer 1 Report

Comments and Suggestions for Authors

This manuscript was edited appropriately according to the reviewer's comments. Thank you for your hard work.

Reviewer 3 Report

Comments and Suggestions for Authors

It could be accepted in the present version.

Comments on the Quality of English Language

English Language is good.